# Endoscopic Surveillance and Treatment of Upper GI Tract Lesions in Patients with Familial Adenomatous Polyposis—A New Perspective on an Old Disease

**DOI:** 10.3390/genes13122329

**Published:** 2022-12-10

**Authors:** Jacek Paszkowski, Paweł Samborski, Marcin Kucharski, Jarosław Cwaliński, Tomasz Banasiewicz, Andrzej Pławski

**Affiliations:** 1Department of General, Endocrinological Surgery and Gastroenterological Oncology, Poznan University of Medical Sciences, 61-701 Poznan, Poland; 2Department of Gastroenterology, Dietetics and Internal Diseases, Poznan University of Medical Sciences, 61-701 Poznan, Poland; 3Institute of Human Genetics, Polish Academy of Sciences, Strzeszyńska 32, 60-479 Poznan, Poland

**Keywords:** familial adenomatous polyposis (FAP), polypectomy, endoscopic ampullectomy, germline mutation

## Abstract

Familial adenomatous polyposis (FAP) is an autosomal dominant disease caused by a germline mutation in the adenomatous polyposis coli (*APC*) gene. Patients with FAP develop up to thousands of colorectal adenomas as well as lesions in the upper GI tract. In FAP, the upper digestive lesions include gastric fundic gland polyps (FGPs), antrum adenomas, duodenal or small intestinal adenomas, and carcinoma. Patients, after colectomy, are still at significant risk for extracolonic malignancies. Advances in endoscope resolution and optical enhancement technologies allow endoscopists to provide assessments of benign and malignant polyps. For this reason, in the past decades, endoscopic resection techniques have become the first line of treatment in patients with polyps in the upper GI, whereby polyps and even early cancers can be successfully cured. In FAP patients, endoscopic ampullectomy appears to be a safe and effective way of treating patients with ampullary tumors. According to current indications, endoscopic retrograde cholangiopancreatography (ERCP) and stenting of the main pancreatic duct follow ampullectomy.

## 1. Introduction

Familial adenomatous polyposis (FAP) is an autosomal dominant hereditary syndrome caused by the mutation of the *APC* gene localized in chromosome 5q21. Although the predominant manifestation of the disease is the presence of numerous colon adenomas, the upper gastrointestinal (GI) tract involvement is well-known in FAP. In FAP patients, there is a significant risk of extracolonic malignancies, especially originating from the small bowel, mainly duodenum, but also the risk of stomach cancer is increased in comparison with that of the population. Duodenal malignancy, desmoid tumors, and CRC are the leading causes of death in these patients. The cumulative risk of duodenal cancer by age of 60 is estimated between 4.5 and 5.3% [1]. Colon endoscopy surveillance and prophylactic colectomy have strongly reduced mortality due to colorectal cancer and have improved the survival of FAP patients, leading to an increased need for surveillance of extracolonic malignancies in the duodenum and stomach. Endoscopic surveillance of FAP patients is a safe and effective way to diagnose precancerous lesions, for example, in the stomach, duodenal, and ampullary adenomas. Moreover, endoscopic resection techniques are the minimally invasive option to treat most of the benign lesions and early cancer in the upper GI tract. However, surveillance recommendations in the upper GI tract are still not completely clear and lack strong evidence.

We searched the PubMed and Google Scholar databases for papers considering endoscopic surveillance and treatment of upper GI tract lesions in patients with familial adenomatous polyposis. The papers included in the analysis were published in the last 30 years, but we emphasized the most recent publications.

## 2. History

The story of FAP endoscopic management is a combination of developing endoscopic instruments, progress in genetic knowledge, and clinical achievements. Endoscopy with a rigid endoscope was first performed and described by Adolf Kusmaul in 1868. The first flexible fiberscope was built almost ninety years later in 1957 by Hirschowitz; in 1970, a gastroscope was presented. Endoscopic polypectomy was first introduced as a novel technique in the early 1970s. The *APC* gene was identified in 1971 and, thereafter, mutations associated with polyposis syndrome were found. The genetic diagnostic test became a standard procedure in the 1990s. Polyps in the stomach in FAP patients were first described and published in 1895, and, almost 10 years later, duodenal lesions were confirmed. Duodenal cancer was first reported in 1962 by Murphy et al. In the year 1989, Spigelman described a staging system of duodenal polyps to stratify the risk of duodenal cancer. The first recommendations for upper GI surveillance were based on the Spigelman classification.

## 3. Stomach Lesions

Patients with FAP frequently develop neoplasms in the upper GI tract. The majority of FAP patients develop gastric polyposis and are at an increased risk for gastric cancer compared with the general population [2,3,4]. According to the National Comprehensive Cancer Network (NCCN) Guidelines: Genetic/Familial High-Risk Assessment: Colorectal, version 1.2022, the estimated lifetime risk of gastric cancer in patients with an inherited APC mutation is 0.1–7.1% compared with 0.8% in the general population, and the average onset is between 52 and 57 years. Cannon. et al. recently (2021), in their publication based on the data of the U.S. polyposis registry, identified gastric cancer as the leading cause of death in FAP and attenuated form of FAP (AFAP) patients after duodenal and ampullary cancers [5]. Screening procedures with the removal of suspicious lesions can prevent the vast majority of carcinomas in these patients. Gastric lesions are common in adult patients with FAP; therefore, the recognition of the types of polyps that can occur in the stomach of FAP patients, early screening with biopsy, and polyp removal are very important for surveillance and prevention.

A variety of different types of polyps can be found in the stomachs of patients with FAP. These types include fundic gland polyps, gastric foveolar-type gastric adenomas, intestinal-type gastric adenomas, pyloric gland adenomas, hyperplastic polyps, and gastric adenocarcinomas.

Fundic gland polyps (FGPs) are the most common type of polyps seen in 40–88% of FAP patients [6,7]. They consist of hyperplasia of the fundic gland, and micro cysts are found in up to 60% of FAP patients. FGPs are typically small (<5 mm), sessile, multiple, asymptomatic, and limited to the stomach (Figure 1). The FGPs associated with FAP differ from sporadic FGPs. FAP-associated FGPs occur in most patients with FAP and show a more equal sex distribution than sporadic FGPs, which are more common in women [8]. In FAP, they are more numerous; therefore, patients with FAP are more likely to have “fundic gland polyposis”. FAP-associated FGPs also occur in children, whereas they are rare in the non-FAP children population. They can be seen in 25–51% of children with FAP undergoing index screening esophagogastroduodenoscopy (EGD) at a mean age of 13 years [7].

Endoscopically, FAP-associated FGPs appear similar to sporadic FGPs, but pathologically, they are dissimilar in that somatic second-hit APC alterations precede morphologic dysplasia in many FAP-associated FGPs, indicating that FGPs are preneoplastic lesions [9]. FGP dysplasia is associated with larger polyp size (>1 cm) and increased severity of duodenal polyposis [10]. The FGPs associated with FAP syndrome differ from sporadic FGPs in their genetic features. They are characterized by alterations in the Wnt signaling pathway genes, particularly *APC* and *CTNNB1* (which encodes β-catenin 1). Somatic, second-hit *APC* alterations precede morphologic dysplasia in many FAP-associated FGPs, showing that FGPs arising in the setting of FAP are neoplastic lesions. Sporadic FGPs without dysplasia have mutations in the gene encoding β-ceratin (*CTNNB1*) but lack *APC* alterations, whereas sporadic FGPs with low-grade dysplasia display APC alterations but usually lack germline mutations in the *CTNNB1* gene [11]. No causal link has been found between *H. pylori* and FAP-associated FGPs.

Adenomas can occur anywhere in the stomach but more commonly occur in the antrum. They are less common than FGPs in patients with FAP. Gastric adenomas in childhood are uncommon in FAP patients, although an increased risk of gastric adenoma in adult FAP patients has been widely recognized. Antral adenomas are usually flat, sessile, and subtle with a villiform red appearance, whereas those in the gastric body and fundus are more polypoid with a pale yellow surface and are therefore difficult to differentiate from FGPs. 

Adenomas localized in the antrum of the stomach should be removed with endoscopic submucosal dissection (ESD) or endoscopic mucosal resection (EMR) techniques when a high degree of suspicion is present during EGD evaluation [12]. Endoscopists performing EGD in FAP patients should have a high degree of suspicion for gastric adenomas, taking frequent biopsies. Virtual endoscopy techniques can be helpful in the detection of stomach lesions, especially flat lesions, and subtle mucosal changes. Lami et al. showed that spectral estimation by Fujinon intelligent color enhancement (FICE) may identify dysplasia and discriminate between adenomatous and nonadenomatous polyps; specifically, that the application of FICE to FAP patients significantly increases the detection of adenomas [13]. This can be extrapolated to virtual endoscopy techniques by other firms such as NBI in Olympus scopes or iScan in Pentax scopes because these endoscopic imagining modalities have been found comparable.

Gastric adenomas were classified by Abraham et al. in 2002 as intestinal-type, containing at least focal goblet cells and/or Paneth cells; and gastric foveolar-type, lined entirely by gastric mucin cells seen on periodic acid-Schiff/Alcian blue staining [14]. The second most common type of polyps seen in FAP patients is the gastric foveolar-type gastric adenoma. However, intestinal-type gastric adenomas are significantly more likely to show high-grade dysplasia, adenocarcinoma within the polyp, intestinal metaplasia in the surrounding mucosa, and gastritis compared with gastric foveolar-type adenomas. Polyps with both intestinal and gastric foveolar differentiation, or “hybrid polyps”, were found to be more aggressive than those with only intestinal-type differentiation [10].

Gastric pyloric gland adenomas (PGAs) are rare epithelial polyps that are more commonly found in autoimmune atrophic gastritis and patients with FAP. They are most often located in the gastric body and show predominance in women. PGAs are clinically significant because they are neoplasms with malignant potential rather than hyperplasia of metaplastic glands. High-grade dysplasia is seen in some cases, and invasive carcinoma is associated with 12–42% of the lesions, depending on the authors’ criteria for carcinoma [11]. The greatest diagnostic challenge with PGAs is distinguishing them from foveolar-type adenomas. PGAs are probably more common than the literature suggests, have a characteristic histologic appearance, and can evolve into infiltrating adenocarcinomas [15]. Histologically, PGAs are composed of closely packed pyloric-type glands with cuboidal to low columnar epithelium showing pale or eosinophilic “ground-glass” cytoplasm with round nuclei, without prominent nucleoli. *GNAS* and *KRAS* germline mutations are present in sporadic PGAs, but also in FAP-associated PGAs despite the different backgrounds in which these lesions arise. In their study, Hashimoto et al. confirmed that FAP-associated PGAs share these distinguishing *GNAS* mutations with sporadic PGAs but that FAP-associated FGPs lack these mutations [16]. As with other types of adenomas, complete excision of PGAs with biopsy of the flat mucosa surrounding the lesion is appropriate in FAP patients, as PGAs often arise in the setting of chronic injury.

Gastric hyperplastic polyps are the most prevalent polyps in regions where *H. pylori* infection is common. In contrast, in western countries, where *H. pylori* infection has a lower prevalence and proton pump inhibitor (PPI) use is common, fundic gland polyps are more prevalent. In their guidelines, the British Society of Gastroenterology suggests that hyperplastic polyps >1 cm, pedunculated morphology, and those causing symptoms (obstruction, bleeding) should be resected. If present, *H. pylori* should be eradicated before re-evaluation. They also recommend that if adenomas or hyperplastic polyps are present, the background mucosa should be endoscopically assessed for gastric atrophy, gastric intestinal metaplasia, *H. pylori*, and synchronous neoplasia [17].

Inflammatory fibroid polyps are uncommon lesions that represent less than 0.1% of all gastric polyps. Following resection, inflammatory fibroid polyps typically do not recur, and surveillance is not recommended [18].

Gastric adenocarcinoma develops from adenoma; it can occur anywhere in the stomach and can be multicentric and metachronous. Shibata et al., in the Japanese population, and Mankaney et al., in the U.S. population, showed that gastric cancer occurs about two decades after colectomy [3,7].

Prior to or concurrent with a gastric cancer diagnosis on endoscopic evaluation, endoscopic features such as carpeting of proximal gastric polyposis, densely concentrated polypoid mounds in the fundus and body of the stomach (1–2 years before cancer diagnosis), and mucosal patches are commonly seen, which make it difficult for surveillance of the stomach for neoplastic lesions [19,20]. Yang et al. in their 2020 guidelines, American Society for Gastrointestinal Endoscopy guideline on the role of endoscopy in familial adenomatous polyposis syndromes, recommend 3- to 6-month surveillance EGD with aggressive polyp sampling and endoscopic debulking of large gastric polyposis mounds because more stage I cancers were found with this protocol. Additionally, they note that mucosal biopsy sampling may not be adequate to assess for malignancy within these thick layers of carpeted polyposis or mounds of gastric polyps and suggest that endoscopic ultrasound (EUS) may help evaluate for an underlying malignancy [7].

Mankaney et al., in their 2022 publication, identified features that, when present, should prompt increased intensity of gastric endoscopic surveillance: gastric white mucosal patches, antral polyps, and family history of gastric cancer, especially in an individual with *APC* pathogenic variant 5′ to codon 1328. In FAP patients, the focus has been on describing the duodenal lesions given their established cancer risk. However, the authors concluded that more care should be taken to carefully examine and describe gastric findings [19].

The American Society for Gastrointestinal Endoscopy (ASGE), in their 2020 guidelines on the role of endoscopy in FAP syndromes, recommends careful evaluation of polyps including FGPs during screening and surveillance endoscopy, with random biopsy sampling and complete resection of polyps >1 cm for evaluation of dysplasia and malignant transformation, particularly in the setting of diffuse gastric polyposis and large gastric mounds. They also recommend that all antral polyps be endoscopically removed, given the high probability of adenoma. The ASGE guidelines (2015) on the role of endoscopy in the management of premalignant and malignant conditions of the stomach state that sampling gastric polyps with forceps can fail to reveal dysplastic components. These guidelines recommend complete snare polypectomy based on size: fundic polyps >10 mm, hyperplastic polyps >5 mm, and all adenomatous polyps [17]. According to the ASGE, surgery should be reserved for patients with FGP and adenomas harboring advanced histologic features who fail endoscopic management [7].

## 4. Duodenal Lesions

Endoscopic surveillance of the GI tract appears to be essential in patients with FAP. Screening colonoscopies in families of FAP patients and prophylactic colectomy in diagnosed patients have importantly reduced the incidence of colorectal cancer [21]. However, the significantly decreased mortality due to lower GI malignancy substantially raised the importance of upper GI tract endoscopy due to the high risk of duodenal adenomatosis and, as a consequence, duodenal cancer incidence. It is calculated to be as high as 4.5–5.3% in the FAP patient population [22,23]. In this way, duodenal cancer is the second leading cause of death in the FAP population after colorectal cancer.

Duodenal adenomas are commonly observed in FAP patients, with an incidence rate reaching 90% [24]. According to the European Society of Gastrointestinal Endoscopy (ESGE) guidelines, small-bowel surveillance should start at the age of 25 in FAP patients [25]. ACG guidelines recommend starting surveillance at the age of 20–30 [26]. Although it was presented more than 30 years ago, the Spigelman classification is still very useful in predicting the risk of duodenal cancer in FAP patients. The frequency of surveillance in the majority of guidelines is based on it; however, the accurate period between examinations differs in various recommendations [27]. It was proven that Spigelman’s classification is not a perfect tool for predicting the progression of dysplasia in duodenal adenomas. One of its flaws is that it is based only on nonampullary adenomatosis extent [22]. Recent studies revealed that ampullary lesions in FAP patients have a strong impact on duodenal cancer risk stratification [28]. In a recent ESGE guideline (2020), this important risk factor was also included.

The rapid development of endoscopes led to the advancement of imaging. There is even evidence that this technological improvement is partially responsible for the increase in the severity of reported duodenal polyps [29]. High-resolution endoscopes enable better evaluation of duodenal adenomas [30]. Very helpful in the assessment of polyps and in predicting their severity stage are virtual chromoendoscopy systems, including narrow-band imaging (NBI, Olympus, Tokyo, Japan), Fujinon intelligent chromoendoscopy (FICE, Fujinon, Tokyo, Japan), and I-Scan (Pentax, Tokyo, Japan) [31]. The underwater technique, in which the duodenal lumen is filled with water to magnify the image, is very useful in some cases as well [32].

In FAP patients, it is essential to carefully assess the ampulla of Vater due to the high occurrence of ampullary adenomas in this population (Figure 2). Standard front-view endoscopy is not a perfect tool to achieve a good image of the periampullary region. The data describing the successful rate of a complete examination of the ampulla of Vater vary depending on the study, but, in recent reports, oscillates between 51 and 54.7% [33,34]. One method to improve visualization of the papilla is cap-assisted endoscopy. A special transparent cap attached to the end of a standard diagnostic esophagogastroscope increases the rate of a complete examination of the ampulla by 95–97% [33,35]. An important advantage of this technique is its simplicity—a transparent cap can be attached during a short break in one procedure, and it does not require special skills from the examining endoscopist. A standard method in use many years before transparent hoods was invented is duodenoscopy. It is still very useful, as it enables the assessment of the periampullary region even in cases where cap-assisted endoscopy is insufficient. However, it requires at least basic ERCP experience from the endoscopist and is less tolerated by patients [36]. Moreover, duodenoscopes are not available in every endoscopy department.

## 5. Endoscopic Treatment

Most of the duodenal lesions can be successfully resected with endoscopic methods [37]. The main goal is to obtain an R0 resection to ensure a good long-term outcome. Thus, the endoscopic technique has to match the size and type of lesion. The histology typical for duodenal lesions in the FAP population is an adenoma, by far the most frequent finding in endoscopic inspections of the duodenum. For practical reasons, duodenal adenomas are divided into nonampullary and ampullary. The endoscopic examination and management of these specific types of adenomas differ.

Nonampullary adenomas are those adenomas found in the duodenum without any connection to the major or minor papilla. They are classified as nonampullary lesions. The recommended management depends on the size and type of the nonampullary lesion. It is generally recommended to assess the type of lesion in Paris classification, especially in the case of adenomas larger than 10 mm. Most of the duodenal polyps in FAP are flat lesions (Paris 0-IIa).

Endoscopic techniques dedicated to nonampullary duodenal adenoma resection are similar to methods used for colonic polypectomies, however with caution on higher adverse event rates. It is a consequence of the rich vascularization of the duodenal wall, thin submucosal tissue, and muscularis propria [37].

In the past, standard management of those polyps was cold-forceps polypectomy of small polyps (<6 mm in diameter) and hot-snare polypectomy or mucosectomy of larger ones. Recently, a cold-snare technique (a polypectomy with the use of a snare and its mechanical ability to cut the tissue without electrosurgical current) became the gold standard for colonic small adenomas as a treatment method. It seems to be useful also in the resection of larger adenomas, in one piece (<10 mm in diameter) as well as in a piecemeal manner (>10 mm in diameter) [38]. Large polyps of a diameter between 10 and 20 mm are usually qualified for hot-snare polypectomy (a polypectomy with the use of a snare attached to electrosurgical current, where the tissue is cut and partly coagulated with the use of electric energy). It is a very effective and relatively safe method to achieve R0 resection in one piece. However, in comparison with the cold-snare technique, hot-snare polypectomy has a higher complication rate, with delayed bleeding and delayed perforation as the most common complications [39]. Duodenal polyps >20 mm is classical exclusion criteria to resect it en block with a snare—it is technically very difficult in the narrow lumen of the duodenum and has a high risk of serious complications. Therefore, snare resection of polyps this size must be performed in fragments, a technique called piecemeal resection. In this clinical situation, the use of electrosurgical current is also associated with a higher risk of delayed bleeding and perforation than a cold snare, similar to the polypectomy of smaller polyps. On the other hand, some data suggest that the cold snare technique is associated with a higher risk of polyp recurrence [40]. Most current studies, however, contradict the increased adenoma recurrence rate after cold-snare polypectomy and conclude that this technique has equal effectiveness and superior safety rate in comparison with hot-snare polypectomy [41,42].

Some accessory techniques might also increase the effectiveness of EMR in the duodenum. One of them is cap-assisted EMR, where the lesion is pulled into a soft cap on the distal end of the gastroscope and then cut by the dedicated snare fitted to the distal end of a cap. In the case of polyps with a diameter suitable to the size of a cap (<15 mm in diameter), cap-assisted EMR can increase a complete resection rate [43]. The other technique useful in duodenal polyp management is underwater EMR. In addition to the already mentioned view enhancement, filling the duodenum with water increases the effectiveness of polyp resection. Water immersion lifts the polypoid tissue increasing the distance from muscularis propria, which results in a higher safety profile and increases the maximal size of the polyp that can be resected en block [44,45]. Moreover, in larger polyps qualified for piece-meal resection, the underwater technique also effectively decreases the complication rate and enables resecting lesions in a reduced number of larger pieces [46,47].

In the case of duodenal adenomas suspected of noninvasive cancer components, achievement of one-piece R0 resection is mandatory. If the size and location of one such lesion have a high risk of noncomplete resection with EMR techniques, endoscopic submucosal dissection (ESD) is a good alternative [48]. However, duodenal ESD has a relatively high risk of complications in comparison with EMR techniques in the duodenum [49]. The ESD technique in the duodenum is also more challenging than in other gastrointestinal locations [50].

Some duodenal adenomas are unsuitable for EMR or ESD resection. One reason can be the difficult location combined with the shape of the polyp. The other possible reason is massive fibrosis and the non lifting characteristic of polyps that often accompanies recurrences. Such polyps can be resected with over-the-scope clip (OTSC)-assisted endoscopic full-thickness resection (EFTR) [51]. This technique, however, is suitable only for lesions smaller than 20 mm because of the size of the cap. 

Ampullary adenomas are a frequent finding in FAP patients. In comparison with surgical options, endoscopic ampullectomy is associated with a much lower risk of complications [52]. Indeed, it has a relatively high recurrence rate reaching 25%; however, the majority of them can be effectively treated with endoscopic methods that lead to the eradication of adenoma [53]. One of the major limitations of endoscopic ampullectomy is the intraductal growth of the tumor. If the infiltration is deeper than 1 cm, it doesn’t meet the criteria of radical treatment [54]. The currently advised technique in ESGE recommendations is hot-snare resection of the ampulla without initial submucosal injection. It is also strongly recommended to insert a short plastic stent into the pancreatic duct at the end of the procedure to prevent pancreatitis, which is the main complication of ampullectomy [25].

Endoscopic treatment methods for duodenal adenomas do not significantly differ in FAP patients in comparison with the rest of the population. However, it is important to notice that precancerous lesions occur in the FAP population at a younger age than spontaneous adenomas, which are typically diagnosed in older patients. This difference has a strong impact on therapeutic decisions, especially in the treatment of benign lesions, in which the risk versus benefit ratio is an important factor.

Surgical treatment should be limited only to cases in which curative resection cannot be achieved with endoscopic techniques. There are two main groups of patients who can benefit from surgical treatment instead of endoscopic resection: those with malignant lesions with deep invasion (deep submucosa or muscularis propria), and those with benign lesions impossible to treat radically with endoscopic methods (for example, ampullary adenomas with intraductal tumor growth deeper than 1 cm) [55].

Moreover, it is very important to consider surgical treatment in patients with advanced Spiegelman score (stage IV) and confirmed high-grade dysplasia, either in biopsy or in the endoscopically resected lesion. Those patients should be regarded as high-risk cancer groups. However, the data about the exact indications to perform surgery in this group are very limited. Recent studies have shown that patients from the high-risk cancer group can significantly benefit from endoscopic treatment, as it can lead to downstaging in Spiegelman score in more than 90% of cases [56]. Long-term surveillance results of those patients are optimistic (more than 70% duodenal surgery-free survival); nonetheless, it appears crucial to obtain more data on large study groups [40].

Regarding the great progress in the knowledge about the genetic basis of FAP, chemoprevention of cancer progression may be a very valuable therapeutic option in addition to the endoscopic and surgical treatment mentioned above. Unfortunately, no medicament has been proven to prevent or delay the progression of malignancy in FAP patients [57]. In recent studies, the mTOR pathway signal inhibitors seem to be one of the most promising groups of agents, but there is still not enough evidence to use them in practice [58].

## 6. Conclusions

Early diagnosis and genetic identification of mutation carriers in FAP families with appropriate endoscopic surveillance have decreased the incidence of malignancy-related deaths during the last decades.More care should be taken to describe not only duodenal but also gastric findings in EGD in patients with FAP, with careful evaluation of polyps, particularly in the setting of gastric polyposis and large gastric mounds.There should be a heightened awareness of the risk of sessile gastric polyps and gastric cancer in patients with FAP.Previous surveillance recommendations might not be completely effective. They still require more data and, as a consequence, need improvement.Technological improvements (HR endoscopy, NBI) delivered very important tools to obtain a diagnosis and make treatment decisions in the precancerous stomach and duodenal lesions in FAP patients.Recently, the endoscopic treatment methods for duodenal lesions have been significantly improved. One of the most important is the cold-snare technique, which is now highly recommended in the resection of lesions <6 mm, but we already have some strong evidence for its effectiveness and safety in piece-meal resection of larger benign duodenal adenomas.

## Figures and Tables

**Figure 1 genes-13-02329-f001:**
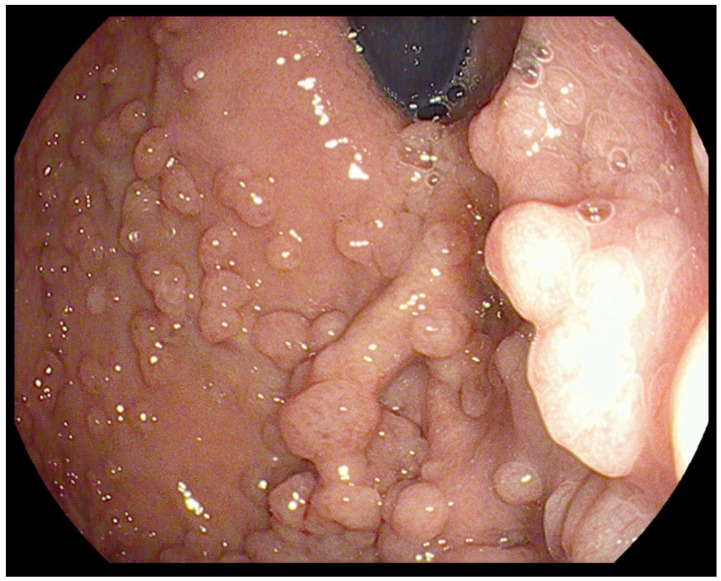
Fundic gland polyps visualized during endoscopy.

**Figure 2 genes-13-02329-f002:**
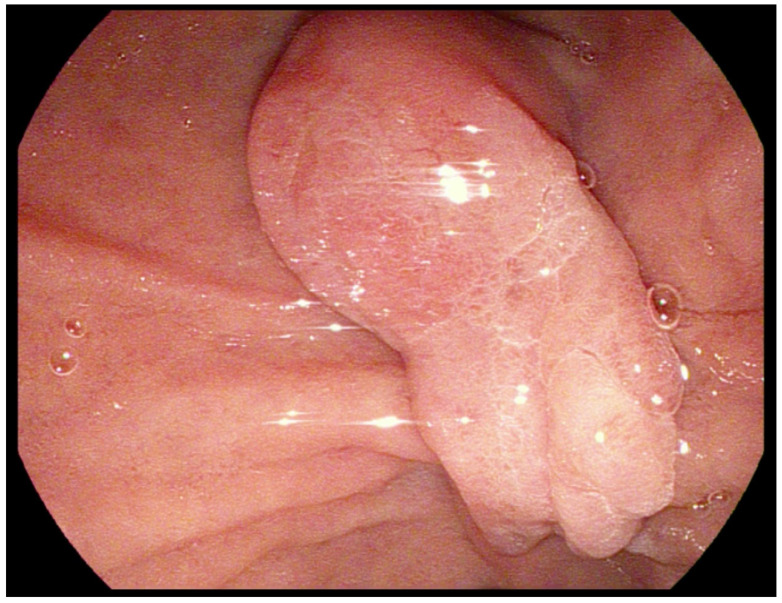
An adenoma of Vater papilla in Standard front-view endoscopy.

## Data Availability

Not applicable.

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
