# Peer review of "Endoscopic Surveillance and Treatment of Upper GI Tract Lesions in Patients with Familial Adenomatous Polyposis—A New Perspective on an Old Disease"

_genes, 2022, doi:10.3390/genes13122329_

Round 1
Reviewer 1 Report
The AA. at the beginnig wrote that they will talk about old and quite question with issues not fully resolved. I not sure that a paer so narrative in the end it is able to bring clarity. Many information but not well organized . The AA have neglected the role of chemoprevention, as well an adequante comment on correlation of genotype and phenotype. GAAPS for exmple. In addition no direct comments on surgery: when ,why and how.
Author Response
The manuscript was improved as follow
In Introduction the section added:
- Colon endoscopy surveillance and prophylactic colectomy have strongly reduced mortality due to colorectal cancer and have improved survival of FAP patients, leading to an increased need for surveillance of extra-colonic malignancies in the duodenum and stomach. Endoscopic surveillance of FAP patients is a safe and effective way to diagnose precancerous lesions, in example stomach, duodenal and ampullary adenomas. Moreover endoscopic resection techniques are minimally invasive option to treat most of benign lesions and early cancer in the upper GI tract. However surveillance recommendations in the upper GI tract are still not completely clear and lacking of strong evidence.” from the History section
From this section removed:
- indications of the journal „The Materials and Methods should be described with sufficient details to allow others to replicate and build on the published results. Please note that the publication of your manuscript implicates that you must make all materials, data, computer code, and protocols associated with the publication available to readers. Please disclose at the submission stage any restrictions on the availability of materials or information. New methods and protocols should be described in detail while well-established methods can be briefly described and appropriately cited. Research manuscripts reporting large datasets that are deposited in a publicly available database should specify where the data have been deposited and provide the relevant accession numbers. If the accession numbers have not yet been obtained at the time of submission, please state that they will be provided during review. They must be provided prior to publication. Interventionary studies involving animals or humans, and other studies that require ethical approval, must list the authority that provided approval and the corresponding ethical approval code.”
In section historywas added
- „In year 1989 Spigelman described a staging system of duodenal polyps to stratify the risk of duodenal cancer. First recommendations for upper GI surveillance were based on Spigelman classification.”
- History section has been changed to Stomach lesions.
In this section added:
- abbreviation „(ASGE)” to „American Society for Gastrointestinal Endoscopy”
- - risk of gastric cancer in patients with the APC mutation was enteres as suggested by the reviewer
- - the text about the mutations in lines 110-113 was improved to make it clearer
- - Recommendations for endoscopic techniques have been moved as suggested from lines 121-132 to 188, where there are other suggestions for endoscopic interventions
- mutations on lines 153-157 also were corrected.
In Endoscopic treatment we added
- „Endoscopic treatment methods of duodenal adenomas do not differ significantly in FAP patients in comparison to the rest of population. It is hovewer important to notice, that pre cancerous lesions occure in FAP population in younger age, than spontanous adenomas, which are typicaly diagnosed in older patients. This difference has a strong impact on therapeutic decisions, espetially in treatment of benign lesions, in which risk versus benefit ratio is an important factor.”
- „Surgical treatment should be limited only to cases, in which curative resection can not be achieved with endoscopic techniques. There are two main groups patients, who can benefit from surgial treatment in instead of endoscopic resection - malgnant lesions with deep invasion (deep submucosa or muscularis propria) and benign lesions imposible to treat radicaly with endoscopic methods (for example ampullary adenomas with intraductal tumor growh deeper than 1 cm) [56].”
in conclusion we odded
- the fourth conclusion „Technological improvement (HR endoscopy, NBI) delivered very important tools to obtain diagnosis and make treatment decisions in pre-cancerous stomach and duodenal lesions in FAP patients.” was interchanged with the fifth conclusion „Previous surveillance recommendations might be not completely effective and still require more data and in consequence need improvement.”
- English was revised by by a long-term scholarship holder in the UK
Reviewer 2 Report
In this review, the authors report the current knowledge on endoscopic surveillance and treatment of lesions of the upper gastrointestinal tract in patients with FAP.
The manuscript is interesting and could be useful in clinical practice. However, in some parts it is not complete, before publication it needs some changes.
The text needs editing for English and grammar; moreover, there are several mistyping.
Title
GI most likely refers to 'gastrointestinal', it should be indicated for greater clarity.
Introduction
This section is inconplete, it should be deepened. The authors should report more information on FAP, and on the various phenotypic forms (profuse, attenuated, intermediate), also because they refer to them later in the text and on the onset of the colorectal cancer in the various forms. In addition, they must indicate that there is a genotype-phenotype correlation, characteristic of these syndrome.
Regarding the APC gene, they must indicate thare are the mutational hotspots related to the severity of the disease. Finally, they must describe more the extracolonic manifestations.
In this regard, the authors can benefit from reading the following articles, that must be added for improving the manuscript:
PMID: 29954149 DOI: 10.3390 / genes9070322
PMID: 31705372 DOI: 10.1007/s11938-019-00251-4
Materials and Methods
- Being a review, should not also be included the ‘Materials and Methods’ section.
- Lines 49-63: these are the indications of the journal, which must be eliminated.
History
This part could be slightly enlarged and described in a more discursive way.
Stomach lesions
- The authors should report the cancer risk in patients with APC mutations.
- Lines 110-113: this part is not clear. Do the authors refer to somatic mutations? or germline mutations, found in peripheral blood? it should be specified.
- Lines 121-132: this part is about endoscopy techniques, it should be moved to section, line 188.
- Lines 153-157: this part is not clear. Do the authors refer to somatic mutations? or germline mutation, found in peripheral blood? it should be specified.
Endoscopic treatement
In this part the authors should specify whether there is a difference of endoscopic treatment in patients with and without APC mutations.
Conclusions
- It should be added an upper common part which introduces the ending points.
- Point 1: this conclusion can be accepted only after the changes made to text.
- Point 5 should be moved after point 3, point 6 after point 4.
Author Response
Changes in the article „Endoscopic surveillance and treatment of upper GI tract lesions in patients with familial adenomatous polyposis - a new perspective on an old disease”.
The manuscript was improved as follow
- Introduction
In this section added:
- word „gastrointestinal” in „upper gastrointestinal (GI) tract”
- word „a” in „a significant risk”
- „mainly duodenum, but also the risk o stomach cancer is increased in comparison to population”
- word „of” in „by age of 60”
- „Colon endoscopy surveillance and prophylactic colectomy have strongly reduced mortality due to colorectal cancer and have improved survival of FAP patients, leading to an increased need for surveillance of extra-colonic malignancies in the duodenum and stomach. Endoscopic surveillance of FAP patients is a safe and effective way to diagnose precancerous lesions, in example stomach, duodenal and ampullary adenomas. Moreover endoscopic resection techniques are minimally invasive option to treat most of benign lesions and early cancer in the upper GI tract. However surveillance recommendations in the upper GI tract are still not completely clear and lacking of strong evidence.” from the History section
- „We searched the PubMed and Google Scholar databases for papers considering endoscopic surveillance and treatment of upper GI tract lesions in patients with familial adenomatous polyposis. The papers included in the analysis were published for last 30 years„ from the Materials and Methods section
- „, but we emphasized the most recent publications.”
From this section removed:
- indications of the journal „The Materials and Methods should be described with sufficient details to allow others to replicate and build on the published results. Please note that the publication of your manuscript implicates that you must make all materials, data, computer code, and protocols associated with the publication available to readers. Please disclose at the submission stage any restrictions on the availability of materials or information. New methods and protocols should be described in detail while well-established methods can be briefly described and appropriately cited. Research manuscripts reporting large datasets that are deposited in a publicly available database should specify where the data have been deposited and provide the relevant accession numbers. If the accession numbers have not yet been obtained at the time of submission, please state that they will be provided during review. They must be provided prior to publication. Interventionary studies involving animals or humans, and other studies that require ethical approval, must list the authority that provided approval and the corresponding ethical approval code.”
- Materials and Methods section has been changed to History.
In this section added:
- words „were found” in „polyposis syndrome were found”
- „In year 1989 Spigelman described a staging system of duodenal polyps to stratify the risk of duodenal cancer. First recommendations for upper GI surveillance were based on Spigelman classification.”
- History section has been changed to Stomach lesions.
In this section added:
- abbreviation „(ASGE)” to „American Society for Gastrointestinal Endoscopy”
- - risk of gastric cancer in patients with the APC mutation was enteres as suggested by the reviewer
- - the text about the mutations in lines 110-113 was improved to make it clearer
- - Recommendations for endoscopic techniques have been moved as suggested from lines 121-132 to 188, where there are other suggestions for endoscopic interventions
- mutations on lines 153-157 also were corrected.
- Duodenal lesions section has been separated out as a new section.
- Endoscopic treatment section has been separated out as a new section.
In this section added:
- „Endoscopic treatment methods of duodenal adenomas do not differ significantly in FAP patients in comparison to the rest of population. It is hovewer important to notice, that pre cancerous lesions occure in FAP population in younger age, than spontanous adenomas, which are typicaly diagnosed in older patients. This difference has a strong impact on therapeutic decisions, espetially in treatment of benign lesions, in which risk versus benefit ratio is an important factor.”
- „Surgical treatment should be limited only to cases, in which curative resection can not be achieved with endoscopic techniques. There are two main groups patients, who can benefit from surgial treatment in instead of endoscopic resection - malgnant lesions with deep invasion (deep submucosa or muscularis propria) and benign lesions imposible to treat radicaly with endoscopic methods (for example ampullary adenomas with intraductal tumor growh deeper than 1 cm) [56].”
- Conclusions
In this section:
- the fourth conclusion „Technological improvement (HR endoscopy, NBI) delivered very important tools to obtain diagnosis and make treatment decisions in pre-cancerous stomach and duodenal lesions in FAP patients.” was interchanged with the fifth conclusion „Previous surveillance recommendations might be not completely effective and still require more data and in consequence need improvement.”
References
In this section added:
- “56. Campos FG. Surgical treatment of familial adenomatous polyposis: dilemmas and current recommendations. World J Gastroenterol. 2014 Nov 28;20(44):16620-9.”
English was revised by by a long-term scholarship holder in the UK
Round 2
Reviewer 1 Report
The AA. certainly improved the paper but some details are still imcorrect as surgical treatment were they do not mention the indication for surgical resention in advanced Spiegman 'stage and when High grade of dysplasia is found. However some mention regarding the chemoprevention should be reported. In addition the bibliogragraphy should be more up-to date
Author Response
English was improved
In the section Endoscopic treatment section was added
Moreover it is very important, to consider surgical treatement in patients with ad-vanced Spiegelman score (stage IV) and confirmed high grade dysplasia, either in biopsy or in endoscopicaly resected lesion. Those patients should be regarded as a high risk of cancer group. However the data about the exact indications to perform surgery in this group are very limited. Especially in recent studies also patients from high risk of cancer group can significantly benefit from endoscopic treatment, as it can lead to downstaging in Spiegelman score in more than 90% of cases [57]. Long-term surveillance results of tho-se patients are optimistic (more than 70% duodenal surgery–free survival), however it appears crutial to get more data on large study groups [58].
Regarding a great progress of knowledge about genetic basis of FAP, chemoprevetion of cancer progression could be a very valuable therapeutic option in addition to endosco-pic and surgical treatment mentioned above. Unfortunately no medicament has been proved, to prevent or delay progression of the malignancy in FAP patients [59]. In recent studies the mTOR pathway signall inhibitors seem to be one of the most promising group of agents, but still there are not enouth evidence, to use them in practice [60].
Lin literature was added
28Nakagawa K, Sho M, Fujishiro M, Kakushima N, Horimatsu T, Okada KI, Iguchi M, Uraoka T, Kato M, Yamamoto Y, Aoyama T, Akahori T, Eguchi H, Kanaji S, Kanetaka K, Kuroda S, Nagakawa Y, Nunobe S, Higuchi R, Fujii T, Yamashita H, Yamada S, Narita Y, Honma Y, Muro K, Ushiku T, Ejima Y, Yamaue H, Kodera Y. Clinical practice guidelines for duodenal cancer 2021. J Gastroenterol. 2022 Dec;57(12):927-941
- Moussata D, Napoleo. n B, Lepilliez V, Klich A, Ecochard R, Lapalus MG et al. Endoscopic treatment of severe duodenal polyposis as an alternative to surgery for patients with familial adenomatous polyposis. Gastrointestinal Endoscopy, Volume 80, Issue 5, 2014.
- Roos VH, Bastiaansen BA, Kallenberg FG, Aelvoet AS, Bossuyt PM, Fockens P et al. Endoscopic management of duodenal adenomas in patients with familial adenomatous polyposis. Gastrointestinal Endoscopy, Volume 93, Issue 2, 2021.
- Kemp Bohan, P.M., Mankaney, G., Vreeland, T.J. et al. Chemoprevention in familial adenomatous polyposis: past, present and future. Familial Cancer 20, 23–33 (2021).
- Rad, E.; Murray, J.T.; Tee, A.R. Oncogenic Signalling through Mechanistic Target of Rapamycin (mTOR): A Driver of Metabolic Transformation and Cancer Progression. Cancers 2018, 10, 5.
